# Distinct responses of frond and root to increasing nutrient availability in a floating clonal plant

Yu Jin[1,2]☯, Qian Zhang[2]☯, Li-Min Zhang[2], Ning-Fei Lei[3], Jin-Song Chen[1]*, Wei Xue[2]*, Fei-Hai Yu[2]

**1** College of Life Science, Sichuan Normal University, Chengdu, China, **2** Institute of Wetland Ecology & Clone Ecology/Zhejiang Provincial Key Laboratory of Plant Evolutionary Ecology and Conservation, Taizhou University, Taizhou, China, **3** Institute of Environment, Chengdu University of Technology, Chengdu, China

☯ These authors contributed equally to this work.
* cjs74@163.com (JSC); x_wei1988@163.com (WX)

## Abstract

Current knowledge on responses of aquatic clonal plants to resource availability is largely based on studies manipulating limited resource levels, which may have failed to capture the "big picture" for aquatic clonal plants in response to resource availability. In a greenhouse experiment, we grew the floating clonal plant *Spirodela polyrhiza* under ten nutrient levels (i.e., 1/64×, 1/32×, 1/16×, 1/8×, 1/4×, 1/2×, 1×, 2×, 4× and 8×full-strength Hoagland solution) and examined their responses in terms of clonal growth, morphology and biomass allocations. The responses of total biomass and number of ramets to nutrient availability were unimodal. A similar pattern was found for frond mass, frond length and frond width, even though area per frond and specific frond area fluctuated greatly in response to nutrient availability. In contrast, the responses of root mass and root length to nutrient availability were U-shaped. Moreover, *S. polyrhiza* invested more to roots under lower nutrient concentrations. These results suggest that nutrient availability may have distinct influences on roots and fronds of the aquatic clonal plant *S. polyrhiza*, resulting in a great influence on the whole *S. polyrhiza* population.

## Introduction

Clonal plants, i.e., those with the ability of clonal growth or asexual reproduction, are widespread in various natural habitats [1–3]. They are also the dominant species in many ecosystems, including grasslands, wetlands and alpine and arctic tundra, where they play a key role in regulating ecosystem functions and stability [2, 4–6]. Clonal plants, via clonal growth, are able to produce offspring ramets (asexual individuals) that have exactly the same genetic information as their mother ramet [3, 7].

Most aquatic plants are capable of clonal growth [8–12]. Aquatic clonal plants are a common component of aquatic communities and play important roles in many aquatic ecosystems [13, 14]. For instance, aquatic clonal plants such as floating and submerged clonal plants can

**Data Availability Statement:** All relevant data are within the manuscript and its Supporting Information files.

**Funding:** This study was supported by the National Natural Science Foundation of China (grant no. 31800341). The funders had no role in study

design, data collection and analysis, decision to publish, or preparation of the manuscript.

**Competing interests:** The authors have declared that no competing interests exist.

reduce turbidity, thereby inhibiting the growth of algae and improving the quality of water [15, 16]. Another example is that oxygen released from the roots of aquatic clonal plants may create an oxidized rhizosphere [17] that facilitates processes contributing to waste degradation [18, 19]. Many aquatic ecosystems are degraded due to, e.g., eutrophication, and a huge amount of efforts have been spent to restore such degraded ecosystems [20–24]. To make better use of aquatic clonal plants for restoration of degraded aquatic ecosystems, we need to assess the detailed responses of their clonal growth, morphology and biomass allocation to a wide range of changes in single environmental factors such as nutrient availability.

Nutrient availability can substantially affect the growth and development of aquatic clonal plants [25, 26]. Excessive nutrient load in aquatic ecosystems can have great consequences on the distribution of aquatic plants, including floating clonal plants [27–31]. For instance, total biomass and ramet number of *Salvinia natans* in high nutrient availability were greater than those in low nutrient availability, and floating mass was mostly higher and submerged mass lower at high than at low nutrient availability [11]. However, in the low ammonia nitrogen ($<8$ mg L$^{-1}$) of water column, an increase in nutrient availability led to increased total biomass and chlorophyll concentration of *Vallisneria natans* [32]. An increase in nutrient availability in the water column leads to a decreased specific root length in *V. natans* [33]. However, plants exhibit slowed growth and altered intrinsic nutrient uptake in a nutrient-poor environment because dry weight production is related to the demand for nutrients [26]. So far, however, studies testing effects of nutrient availability on clonal growth, morphology and biomass allocation of aquatic clonal plants, especially floating clonal plants, have mostly included only a few (e.g., 2–4) nutrient levels [12, 34–38]. To better understand their responses, therefore, we need to test their responses to a wider range of nutrient levels.

We grew the floating clonal plant *Spirodela polyrhiza* in ten concentrations of Hoagland solution and evaluated the effects of nutrient availability on clonal growth, morphology and biomass allocation. Specifically, we addressed the following questions. (1) How does nutrient availability affect clonal growth of *S. polyrhiza* as measured by biomass, ramet production and total frond area? (2) How does nutrient availability influence clonal morphology of *S. polyrhiza* as measured by frond length, frond width, the longest root length, area per frond and specific frond area? (3) How does nutrient availability impact root to shoot ratio of *S. polyrhiza*?

## Materials and methods

### Species and sampling

*Spirodela polyrhiza* (L.) Schleiden is a perennial, floating, clonal plant of Lemnaceae (duckweed family). It has the simplest morphology among flowering plants [39]. The species rarely flowers and mainly reproduces vegetatively [40, 41]. A ramet (i.e., asexual individual) of *S. polyrhiza* commonly consists of one or two fronds and some roots [12]. When environmental conditions are favorable, a parent ramet can produce offspring ramets that are connected to it by a stipe at their early stage of development [39]. Offspring ramets can be detached from the parent ramet and become completely independent due to aging or disturbance [12]. Each frond of *S. polyrhiza* is 5–10 mm long and 3–8 mm wide. It is flat and obovate, with a green surface towards the air and purple back towards the water. This species is widely distributed across the world, and typically found in eutrophic freshwater systems, such as slow-moving streams, ditches, and shallow pools [12, 41].

Plants of *S. polyrhiza* were collected on June 2, 2018 from a slow-moving stream (28˚3′N, 121˚21′E) in Jiaojiang District, Taizhou City, Zhejiang Province, China. Plants were transported to the greenhouse at the Jiaojiang campus of Taizhou University in Jiaojiang District, Taizhou City, Zhejiang Province, China. The plants were washed several times with double

distilled water and rinsed with 0.01 M NaClO for 30 s to reduce microbial and algal growth [42]. Before the experiment, the plants were propagated vegetatively in plastic tanks with 10% Hoagland solution.

## Experimental design

The experiment started on June 19, 2018. Healthy ramets of *S. polyrhiza* with uniform size and two fronds were selected and grown in 250 mL containers (9.5 cm in diameter and 6.5 cm in height). Each container initially contained two ramets of *S. polyrhiza*. Plants were subjected to ten levels of full-strength Hoagland nutrient solution ($1/64\times$, $1/32\times$, $1/16\times$, $1/8\times$, $1/4\times$, $1/2\times$, $1\times$, $2\times$, $4\times$ and $8\times$), with 12 replicates for each treatment. The composition of $1\times$full-strength Hoagland solution included: 0.945 g $L^{-1}$ $Ca(NO_3)_2\cdot4H_2O$, 0.506 g $L^{-1}$ $KNO_3$, 0.08 g $L^{-1}$ $NH_4NO_3$, 0.136 g $L^{-1}$ $KH_2PO_4$, 0.493 g $L^{-1}$ $MgSO_4$, 0.0139 g $L^{-1}$ $FeSO_4\cdot7H_2O$, 0.01865 g $L^{-1}$ $EDTA\cdot2Na$, 0.00415 g $L^{-1}$ KI, 0.031 g $L^{-1}$ $H_3BO_3$, 0.1115 g $L^{-1}$ $MnSO_4$, 0.043 g $L^{-1}$ ZnSO4, 0.00125 g $L^{-1}$ $Na_2MoO_4$, 0.000125 g $L^{-1}$ $CuSO_4$, and 0.000125 g $L^{-1}$ $CoCl_2$. Each container was filled with 200 mL of nutrient solution. Every four days, nutrient solution was replaced in all containers. Containers were placed completely randomly. After 14 days, on July 3, 2018, plants were harvested. The experiment ran within a relative short period because the surface of the solution was already fully covered with the ramets of *S. polyrhiza* in some containers where intraspecific competition would occur if the experiment ran for a longer time. During the experiment, the mean air temperature was 24.3˚C and the mean relative humidity was 81.3% in the greenhouse, as measured hourly using temperature loggers (iButton DS1923; Maxim Integrated Products, San Jose, CA, USA).

## Measurements

At harvest, we counted ramets and fronds of *S. polyrhiza* in each container. We randomly selected five ramets in each container, and measured their frond length, frond width and longest root length. Total area of all fronds in each container was scanned and measured with ImageJ 2006 (Bethesda, MD, USA). All ramets in each container were separated into roots and fronds, dried at 75˚C for 24 h, and weighed.

## Data analysis

We calculated area per frond (total frond area/number of fronds), root to shoot ratio (total root mass/total frond mass) and specific frond area (total frond area/total frond mass). We also obtained mean frond length, mean frond width and mean longest root length based on the measures on the five ramets in each container. The data of fourteen containers were excluded from the analyses due to missing values.

One-way ANOVA was used to evaluate the effects of nutrient availability on clonal growth (total mass, frond mass, root mass, number of ramets and total frond area), morphology (frond length, frond width, longest root length, area per frond and specific frond area) and biomass allocation (root to shoot ratio). Before analysis, data on number of ramets, total frond area, frond mass, total mass, and SLA were transformed to logarithmic, data on root mass were transformed to square root, data on longest root length was transformed to trigonometric. Other data required no transformation to meet requirements for homoscedasticity and normality; figures showed untransformed data. Duncan's multiple range test was used to compare mean values among treatments. All analyses were conducted using SPSS 22.0 (IBM, Armonk, NY, USA).

## Results

### Effects of nutrient availability on clonal growth

Nutrient availability highly significantly affected biomass ($F_{9,96}$ = 2.8–48.9, all $P < 0.01$), number of ramets ($F_{9,96}$ = 94.7, $P < 0.001$) and total frond area ($F_{9,96}$ = 34.2, $P < 0.001$) of *S. polyrhiza* (Figs 1 and 2). Total mass and frond mass of *S. polyrhiza* initially increased and then declined as nutrient availability increased, and the maximum value was observed at 4×full-strength Hoagland solution (Fig 1A and 1B). Root mass initially decreased, then increased and finally decreased again, showing the greatest value at 4×full-strength Hoagland solution, smallest values at 1/8× and 1/4×full-strength Hoagland solution and intermediate values at other concentrations (Fig 1C). Number of ramets and total frond area showed the similar patterns as total mass and leaf mass (Fig 2).

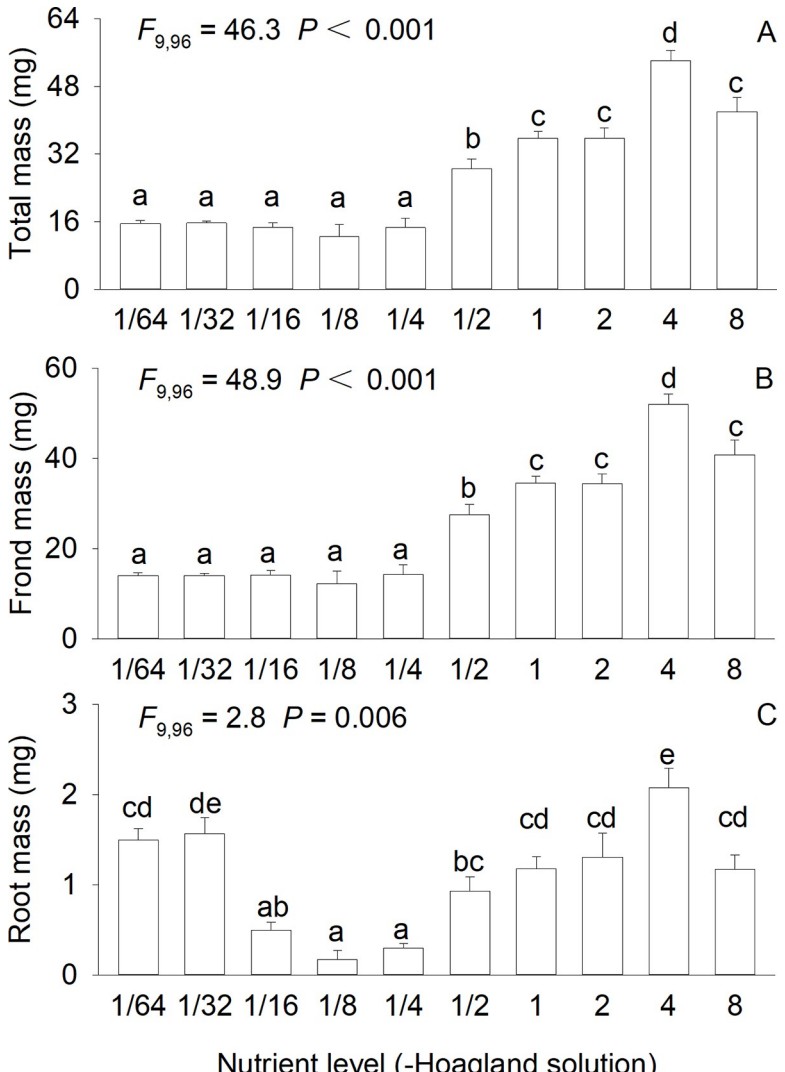

**Fig 1.** Total mass (A), frond mass (B), and root mass (C) of *Spirodela polyrhiza* grown in ten concentrations of Hoagland nutrient solution. Bars and vertical lines indicate means and SE (n = 12). F-statistics, df and P-values of one-way ANOVA for the effect of nutrient level are also given. Bars sharing the same letters are not different at $P = 0.05$.

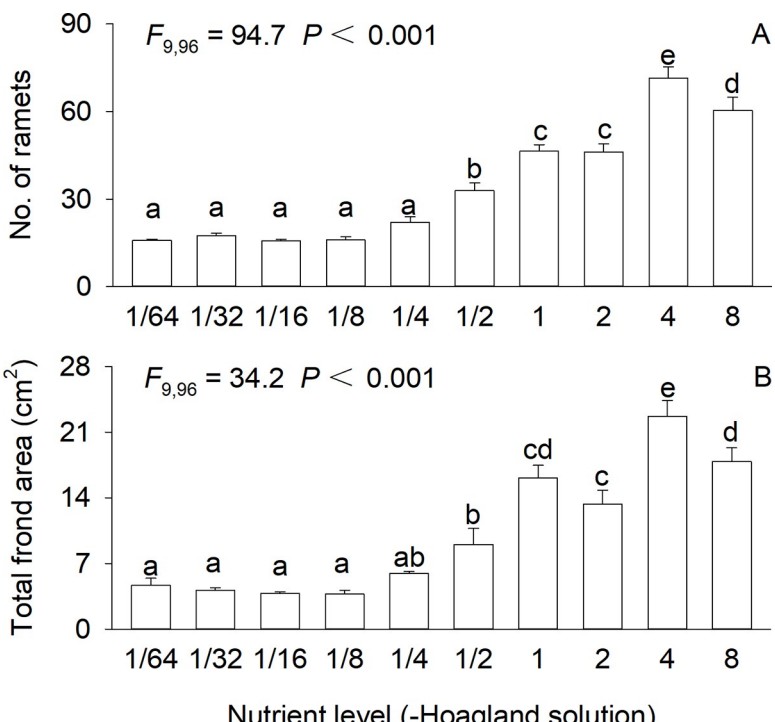

**Fig 2.** Ramet number (A) and total frond area (B) of *Spirodela polyrhiza* grown in ten concentrations of Hoagland nutrient solution. Bars and vertical lines indicate means and SE (n = 12). F-statistics, df and P-values of one-way ANOVA for the effects of nutrient level are also given. Bars sharing the same letters are not different at $P = 0.05$.

## Effects of nutrient levels on biomass allocation

Nutrient availability highly significantly affected root to shoot ratio of *S. polyrhiza* ($F_{9,96}$ = 24.5, $P < 0.001$; Fig 3). Root to shoot ratio of *S. polyrhiza* at 1/64× and 1/32×full-strength Hoagland solution did not differ significantly, but was much larger than that at all other eight concentrations (Fig 3). Root to shoot ratio of *S. polyrhiza* also also greater in 1/16×, 2× and 4× full-strength Hoagland solution than in 1/8× full-strength Hoagland solution (Fig 3).

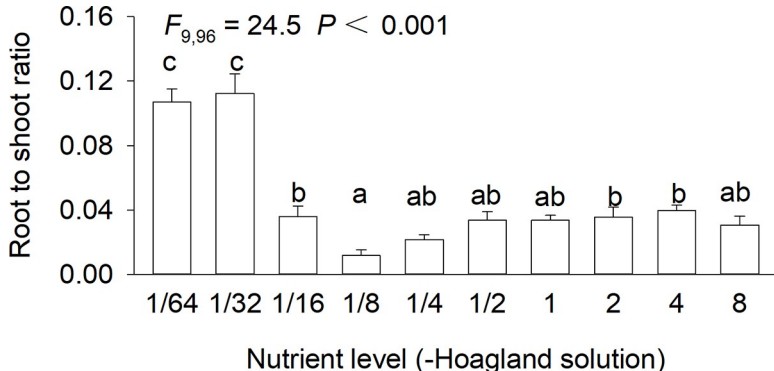

**Fig 3. Root to shoot ratio of *Spirodela polyrhiza* grown in ten concentrations of Hoagland nutrient solution.** Bars and vertical lines indicate means and SE (n = 12). F-statistics, df and P-values of one-way ANOVA for the effects of nutrient level are also given. Bars sharing the same letters are not different at $P = 0.05$.

### Effects of nutrient availability on clonal morphology

Clonal morphology of *S. polyrhiza* was significantly affected by nutrient availability ($F_{9,96}$ = 2.1–12.6, all $P < 0.05$; Fig 4). Both frond length and frond width first gradually increased with increasing nutrient level, maximized at 4×full-strength Hoagland solution, and then decreased at 8×full-strength Hoagland solution (Fig 4A and 4B). With increasing nutrient availability, length of the longest root first decreased sharply, minimized at 1/8×full-strength Hoagland solution, and then increased, with the greatest values occurring at the two lowest concentrations (1/64× and 1/32×) of Hoagland solution (Fig 4E). Neither specific frond area nor area per frond showed a clear pattern, and they fluctuated with increasing nutrient availability (Fig 4C and 4D).

## Discussion

Our results showed that the responses of total biomass and number of ramets to increasing nutrient availability were hump-shaped. Similar patterns were found for frond mass, frond length and frond width, despite that no clear pattern was found for area per frond and specific frond area. In contrast, the responses of root mass and root length to increasing nutrient availability were U-shaped. These results indicated that fronds and roots of the same clonal plant may had different response strategies to increasing nutrient availability, which may have largely determined the performance of the whole *S. polyrhiza* population.

### Hump-shaped growth patterns in response to nutrient availability

Our results showed an overall hump-shaped growth pattern of *S. polyrhiza* in response to increasing nutrient availability, in agreement with many theoretic and experimental studies [41, 43–46]. The initial increase of the biomass was likely due to the very low water nutrient supply as plant production is generally positively correlated with the demand for nutrients [26, 41, 47]. The declined growth at higher levels of nutrient availability can be explained by four reasons in our study. First, the biomass gained from the newly ramets did not level off the biomass lost from the older ramets, which may have resulted in the decreased growth at the whole pot level. Second, intraspecific competition increased as the growth of the *S. polyrhiza* population, which may have led to the decreased growth at higher nutrient levels [8, 10, 32, 48, 49]. Third, it was also likely that ammonia nitrogen and heavy metal elements in Hoagland's solutions of high concentrations may have reduced the growth of *S. polyrhiza*, which has also been found in other species [32, 44, 50, 51]. Moreover, we cannot rule out the possibility that the complex interactions among elements (ions) in Hoagland's solutions of high concentrations played a role in shaping growth pattern of *S. polyrhiza* [52–54].

### Contrasting responses of frond and root to nutrient availability

Fronds are the main photosynthetic organs of *S. polyrhiza*, and therefore their morphology may have great influences on plant growth [12, 38, 55–57]. We observed that total frond area, frond length and width and frond mass of *S. polyrhiza* increased with increasing nutrient availability but decreased afterwards. The response of frond to nutrient availability may have largely determined the growth pattern of *S. polyrhiza* in response to nutrient availability. In general, a greater specific leaf area would be expected under shaded or crowded environments where light is limited [58–61]. However, we did not observe a clear pattern for specific frond area in response to nutrient levels, despite that the population density in terms of total biomass and number of ramets changed a lot with the applied nutrient levels. This result indicated that specific frond area may not be a good predictor for light competition driven by nutrient-induced population growth.

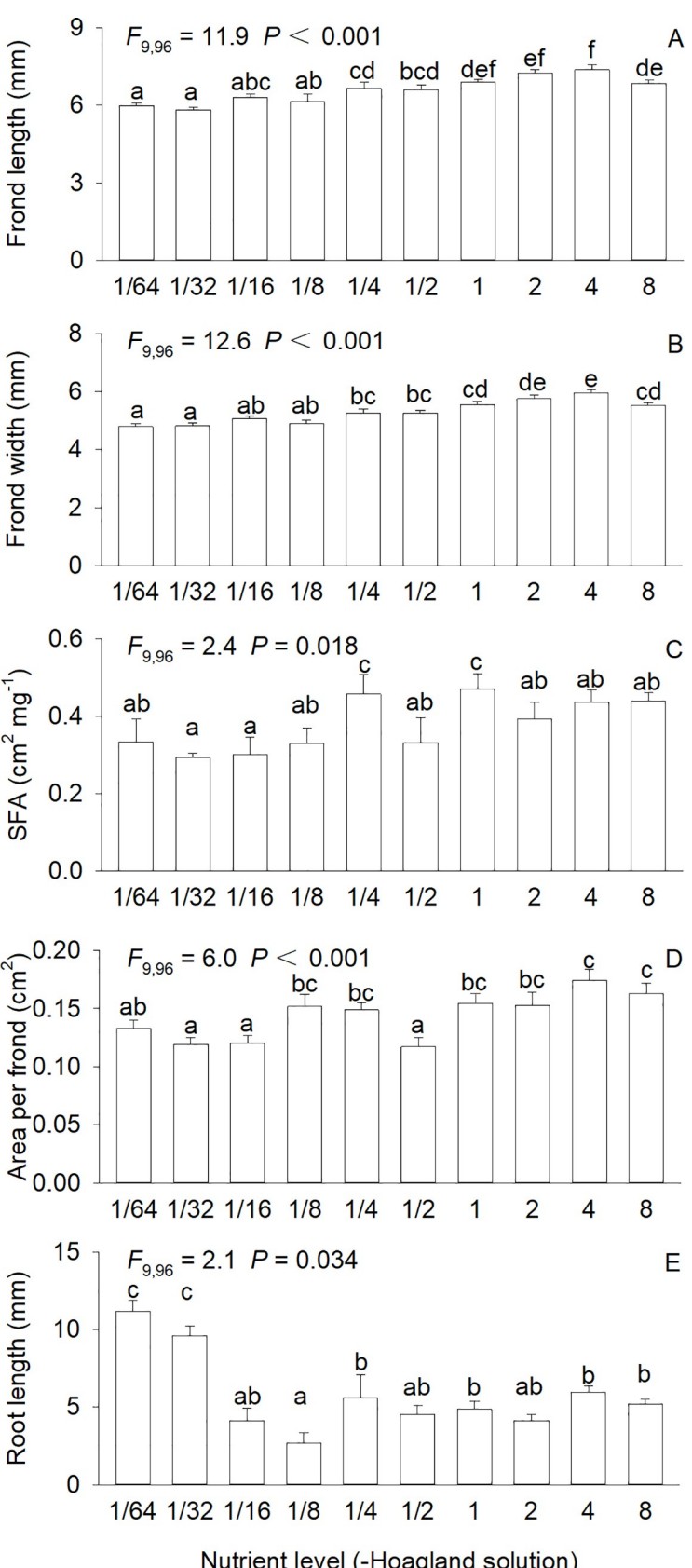

**Fig 4.** Frond length (A), frond width (B), specific frond area (SFA, C), area per frond (D), and longest root length (E) of *Spirodela polyrhiza* grown in ten concentrations of Hoagland nutrient solution. Bars and vertical lines indicate means and SE (n = 12). F-statistics, df, and P-values of one-way ANOVA for the effect of nutrient levels are also given. Bars sharing the same letters are not different at *P* = 0.05.

In contrast, a U-shaped pattern was observed in root length and root mass in response to increasing nutrient levels. This is likely because, at lower level of nutrient concentrations, plants may need to invest more to roots in order to uptake more nutrients [43, 62–64], which could explain the greater root length and root mass at lower nutrient concentration treatments in the present study [65]. However, we also observed a greater root length and root mass at higher nutrient concentrations. This is likely because at higher levels of nutrient concentration, plants may suffer from strong intraspecific competitions due to rapidly clonal growth. In this case, a longer and bigger root would be more beneficial for the plants to obtain resources in order to fight against their intraspecific competitors [11, 43, 64, 66].

## Trade-offs between root and shoot growth in response to nutrient availability

Plants can respond to varying resources through changing biomass allocations to above- and below-ground organs [67, 68]. In general, many terrestrial plants would allocate more biomass to below-ground organs under low-nutrient soils [66, 69–71], and this is also frequently observed for many aquatic macrophytes [72, 73]. We also observed a greater root to shoot ratio at lower nutrient concentrations in the floating clonal plant *S. polyrhiza*. However, we do not know whether it is a common strategy in floating clonal plants in response to increasing nutrient availability, as clonal plants are characterized by highly plasticity in morphology [74–76]. Therefore, more studies on other floating clonal plants are required to generalize our findings.

## Conclusions

We conclude that the floating clonal plant *S. polyrhiza* varied in terms of growth and morphology in response to increasing nutrient availability. In general, *S. polyrhiza* showed a hump-shaped growth pattern as increased nutrient availability, but shoots and roots of *S. polyrhiza* differed in their responses to the nutrient availability. Our results have important implications for the control of eutrophication which is common in natural ecosystem [77–80]. However, one should be noted that in this study we only used one floating clonal plant and our experiment only ran for a short period, we do not know how *S. polyrhiza* and many other floating clonal plants may vary in their responses to increasing nutrient availability in the long run. Therefore, to generalize our findings, future research should focus on the long-term effects of changing nutrients or other biological factors on various floating clonal plants.

## Supporting information

**S1 Data.**
(XLSX)

## Acknowledgments

We gratefully thank Si-Mei Yao for her help in the harvest experiment.

## Author Contributions

**Conceptualization:** Yu Jin, Jin-Song Chen, Wei Xue, Fei-Hai Yu.

**Data curation:** Yu Jin, Li-Min Zhang, Ning-Fei Lei.

**Formal analysis:** Yu Jin, Qian Zhang.

**Funding acquisition:** Qian Zhang.

**Methodology:** Yu Jin, Li-Min Zhang, Wei Xue.

**Software:** Li-Min Zhang.

**Writing – original draft:** Yu Jin.

**Writing – review & editing:** Wei Xue, Fei-Hai Yu.

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
