## [Decision Letter · Decision Letter 0]

19 Aug 2021

PONE-D-21-23023

Distinct responses of frond and root to increasing nutrient availability in a floating clonal plant

PLOS ONE

Dear Dr. Jin,

Thank you for submitting your manuscript to PLOS ONE. After careful consideration, we feel that it has merit but does not fully meet PLOS ONE’s publication criteria as it currently stands. Therefore, we invite you to submit a revised version of the manuscript that addresses the points raised during the review process.

The study is interesting while the manuscript has some problems as suggested by the reviewers. The authors should respond to the comments of the reviewers one by one and revise the manuscript accordingly. The revised manuscript would be sent to the reviewers for further reviewing.

We look forward to receiving your revised manuscript.

Kind regards,

Jian Liu

Academic Editor

PLOS ONE

Journal Requirements:

Reviewers' comments:

Reviewer's Responses to Questions

**Comments to the Author**

1. Is the manuscript technically sound, and do the data support the conclusions?

Reviewer #1: Yes

Reviewer #2: Yes

2. Has the statistical analysis been performed appropriately and rigorously? 

Reviewer #1: Yes

Reviewer #2: Yes

3. Have the authors made all data underlying the findings in their manuscript fully available?

Reviewer #1: Yes

Reviewer #2: Yes

4. Is the manuscript presented in an intelligible fashion and written in standard English?

Reviewer #1: Yes

Reviewer #2: Yes

5. Review Comments to the Author

Reviewer #1: It’s an interesting work. The manuscript read fluently. I only have a few small questions.

1. The authors used 10 levels of Hoagland’s solution. There are some different formulas of Hoagland’s solution. Could the authors show the exact solution composition?

2. Effects of nutrients on plants’ growth have been linked to compositions and concentrations. For Egeria densa, ammonia nitrogen blow 1 mg L-1 could promote growth of plants, but ammonia nitrogen exceeding 10 mg L-1 might produce oxidative stress to plants and inhibit the growth. For Vallisneria natans, the thresholds of promotion and inhibition effects of ammonia nitrogen are respectively lower than 0.6 mg L-1 and higher than 1 mg L-1. Ammonia nitrogen in high concentrations of Hoagland’s solutions might make an adverse effect on growth, which should be considered and discussed.

3. There has been study showing that copper exceeding 1 μmol L-1 significantly inhibits growth of Spirodela polyrhiza. Mn exceeding certain concentration also has negative effect on growth of Spirodela polyrhiza. It has been proved by many studies that low concentration of heavy metal could promote growth. Heavy metal elements come from high concentrations of Hoagland’s solutions might also make adverse effects on growth. Some studies have also reported that heavy metal pollution decreased root to shoot ratio. Effects of heavy metal should be considered.

4. Different elements (ions) have antagonistic or synergistic effects with each other. An antagonistic (synergistic) relationship between elements (ions) at certain concentration may change into a synergistic (antagonistic) relationship at another concentration. For example, Ca2+ and K+ have antagonistic and competitive effects at normal concentration, while Ca2+ can promote the uptake of K+ by plants at very low concentration. The relationship between elements (ions) is complex due to composition of Hoagland’s solution, especially at high concentrations. This may have something with the hump-shaped and U-shaped changes.

It is suggested the authors should dive deeper into the experimental phenomenon.

Reviewer #2: The manuscript reported a case study that tested the effects of nutrient availability on clonal performance (growth, morphology and biomass allocation) of a floating clonal plant Spirodela polyrhiza. The novelty of the paper is to explore plant responses to a wide gradient of nutrient availability (1/64 to 8 × full-strength Hoagland solution), which have rarely been examined in previously published studies. The experimental design is clear, and the content was well edited.

However, when I checked the raw data in the supplementary material, there is one serious problem. It is not logical that the values of root mass and/or root to shoot ratio of S. polyrhiza are treated as zero when root length has non-zero values. The reverse situation is also not logical.

If some of data are missing, they should be treated as null values and excluded from the analyses. I suggest that the data should be re-analyzed and re-discussed after they are carefully checked.

Other minor problems were listed below.

Line 17, delete “or”.

Line 49, replace “flower” with “flowers”.

Line 69, replace “sizes” with “size”.

Line 75, delete “of time”.

Line 101, replace “show” with “showed”.

Line 111, It appears to be better to use the phrase “4 × full-strength Hoagland solution”, compared to the used one “4-Hoagland solution”, when describing the levels of nutrient availability.

Line 172, use “reasons”, instead of “mechanisms”.

Line 176 – 178, In your study, the nutrient solutions were refreshed with the four-day interval, so it seems to be sufficient for plant growth during the entire experimental period.

Line 216, delete “a lot”.

Line229, “master student”?

Line 230 – 438, the reference formats and some spelling errors should be carefully checked and corrected.

For instance, spelling errors: “ence” in line 315; “thr” in 362. Format errors: line 287-288; line 343, line 355-356; line 359; line 364-366; line 367-369; line 378; line 423.

6. PLOS authors have the option to publish the peer review history of their article (what does this mean?). If published, this will include your full peer review and any attached files.

Reviewer #1: No

Reviewer #2: No

---

## [Author Response · Author response to Decision Letter 0]

1 Sep 2021

Reviewer #1: It’s an interesting work. The manuscript read fluently. I only have a few small questions.

Response: Thanks.

The authors used 10 levels of Hoagland’s solution. There are some different formulas of Hoagland’s solution. Could the authors show the exact solution composition?

Response: We have added the exact solution composition in the revised version (lines 94 – 98).

Effects of nutrients on plants’ growth have been linked to compositions and concentrations. For Egeria densa, ammonia nitrogen blow 1 mg L-1 could promote growth of plants, but ammonia nitrogen exceeding 10 mg L-1 might produce oxidative stress to plants and inhibit the growth. For Vallisneria natans, the thresholds of promotion and inhibition effects of ammonia nitrogen are respectively lower than 0.6 mg L-1 and higher than 1 mg L-1. Ammonia nitrogen in high concentrations of Hoagland’s solutions might make an adverse effect on growth, which should be considered and discussed.

Response: Nice point, thanks. We have addressed this issue in the discussion section. (Lines 224 – 227).

There has been study showing that copper exceeding 1 μmol L-1 significantly inhibits growth of Spirodela polyrhiza. Mn exceeding certain concentration also has negative effect on growth of Spirodela polyrhiza. It has been proved by many studies that low concentration of heavy metal could promote growth. Heavy metal elements come from high concentrations of Hoagland’s solutions might also make adverse effects on growth. Some studies have also reported that heavy metal pollution decreased root to shoot ratio. Effects of heavy metal should be considered.

Response: We agree and we have discussed it in the revised version (Lines 224 – 227). 

Different elements (ions) have antagonistic or synergistic effects with each other. An antagonistic (synergistic) relationship between elements (ions) at certain concentration may change into a synergistic (antagonistic) relationship at another concentration. For example, Ca2+ and K+ have antagonistic and competitive effects at normal concentration, while Ca2+ can promote the uptake of K+ by plants at very low concentration. The relationship between elements (ions) is complex due to composition of Hoagland’s solution, especially at high concentrations. This may have something with the hump-shaped and U-shaped changes.

It is suggested the authors should dive deeper into the experimental phenomenon.

Response: Good to know, thanks. In the revised version, we also discussed the potential relationship between different elements in driving the responses of Spirodela polyrhiza to nutrient availability (Lines 227 – 229). 

Reviewer #2: The manuscript reported a case study that tested the effects of nutrient availability on clonal performance (growth, morphology and biomass allocation) of a floating clonal plant Spirodela polyrhiza. The novelty of the paper is to explore plant responses to a wide gradient of nutrient availability (1/64 to 8 × full-strength Hoagland solution), which have rarely been examined in previously published studies. The experimental design is clear, and the content was well edited.

Response: Thanks.

However, when I checked the raw data in the supplementary material, there is one serious problem. It is not logical that the values of root mass and/or root to shoot ratio of S. polyrhiza are treated as zero when root length has non-zero values. The reverse situation is also not logical.

If some of data are missing, they should be treated as null values and excluded from the analyses. I suggest that the data should be re-analyzed and re-discussed after they are carefully checked.

Response: We agree and we have reanalyzed our data by excluding the null values, which did not change our results. We have updated the results in the revised version (Lines139 – 145, 165 – 170 and 178 – 181).

Specific comments:

Introduction

Line 17, delete “or”.

Response: Deleted (Line 36).

Materials and methods

Line 49, replace “flower” with “flowers”.

Response: Replaced (Line 69).

Line 69, replace “sizes” with “size”.

Response: Replaced (Line 90).

Line 75, delete “of time”.

Response: Deleted (Line 101).

Line 101, replace “show” with “showed”.

Response: Replaced (Line 131).

Results

Line 111, It appears to be better to use the phrase “4 × full-strength Hoagland solution”, compared to the used one “4-Hoagland solution”, when describing the levels of nutrient availability.

Response: Thanks, we replaced “4-Hoagland solution” with “4 × full-strength Hoagland solution” (Lines 143 – 145). We also changed the phrase throughout the whole manuscript (Lines 6 – 7, 92 – 93, 166 – 170 and 180 – 184).

Discussion

Line 172, use “reasons”, instead of “mechanisms”.

Response: Changed (Line 220).

Line 176 – 178, In your study, the nutrient solutions were refreshed with the four-day interval, so it seems to be sufficient for plant growth during the entire experimental period.

Response: Yes, we agree and we have removed this explanation.

Conclusions

Line 216, delete “a lot”.

Response: Deleted (Line 284).

Line229, “master student”?

Response: Deleted “Master” (Line 272).

References

Line 230 – 438, the reference formats and some spelling errors should be carefully checked and corrected.

For instance, spelling errors: “ence” in line 315; “thr” in 362. Format errors: line 287-288; line 343, line 355-356; line 359; line 364-366; line 367-369; line 378; line 423.

Response: We have checked and reformatted our references in the revised version (Lines 288 –507).

---

## [Decision Letter · Decision Letter 1]

23 Sep 2021

Distinct responses of frond and root to increasing nutrient availability in a floating clonal plant

PONE-D-21-23023R1

Dear Dr. Jin,

We’re pleased to inform you that your manuscript has been judged scientifically suitable for publication and will be formally accepted for publication once it meets all outstanding technical requirements.

Kind regards,

Jian Liu

Academic Editor

PLOS ONE

Additional Editor Comments (optional):

Reviewers' comments:

Reviewer's Responses to Questions

**Comments to the Author**

1. If the authors have adequately addressed your comments raised in a previous round of review and you feel that this manuscript is now acceptable for publication, you may indicate that here to bypass the “Comments to the Author” section, enter your conflict of interest statement in the “Confidential to Editor” section, and submit your "Accept" recommendation.

Reviewer #1: All comments have been addressed

Reviewer #2: All comments have been addressed

2. Is the manuscript technically sound, and do the data support the conclusions?

Reviewer #1: Yes

Reviewer #2: Yes

3. Has the statistical analysis been performed appropriately and rigorously? 

Reviewer #1: Yes

Reviewer #2: Yes

4. Have the authors made all data underlying the findings in their manuscript fully available?

Reviewer #1: Yes

Reviewer #2: Yes

5. Is the manuscript presented in an intelligible fashion and written in standard English?

Reviewer #1: Yes

Reviewer #2: Yes

6. Review Comments to the Author

Reviewer #1: The manuscript has improved. I have no more comments on the manuscript itself. However, the line numbers in responses are wrong. The author said questions were answered and discussed in Lines 94-98, 224-229. I spend some time in checking, and guess I find the right positions. The author should be more careful with writing.

Reviewer #2: (No Response)

7. PLOS authors have the option to publish the peer review history of their article (what does this mean?). If published, this will include your full peer review and any attached files.

Reviewer #1: No

Reviewer #2: No

---

## [Editor Report · Acceptance letter]

27 Sep 2021

PONE-D-21-23023R1 

Distinct responses of frond and root to increasing nutrient availability in a floating clonal plant 

Dear Dr. Jin:

I'm pleased to inform you that your manuscript has been deemed suitable for publication in PLOS ONE. Congratulations! Your manuscript is now with our production department. 

Kind regards, 

on behalf of

Dr. Jian Liu 

Academic Editor

PLOS ONE